

# Commercial utilities and future perspective of nanomedicines

Rishabha Malviya[1], Shivkanya Fuloria[2], Swati Verma[1], Vetriselvan Subramaniyan[3], Kathiresan V. Sathasivam[4], Vinoth Kumarasamy[3], Darnal Hari Kumar[5], Shalini Vellasamy[3], Dhanalekshmi Unnikrishnan Meenakshi[6], Shikha Yadav[1], Akanksha Sharma[1] and Neeraj Kumar Fuloria[2]

[1] Department of Pharmacy, SMAS, Galgotias University, Greater Noida, Uttar Pradesh, India
[2] Faculty of Pharmacy & Centre of Excellence for Biomaterials Engineering, AIMST University, Bedong, Kedah, Malaysia
[3] Faculty of Medicine, Bioscience and Nursing, MAHSA University, Bandar Saujana Putra, Selangor, Malaysia
[4] Faculty of Applied Science & Centre of Excellence for Biomaterials Engineering, AIMST University, Bedong, Kedah, Malaysia
[5] Jeffrey Cheah Cheshire School of Medicine & Health Sciences, Monash University, Selangor, Malaysia
[6] College of Pharmacy, University of Science and Technology, Muscat, Oman

Corresponding author
Neeraj Kumar Fuloria,
neerajkumar@aimst.edu.my

## ABSTRACT

The present review aims to describe the commercial utilities and future perspectives of nanomedicines. Nanomedicines are intended to increase precision medicine and decrease the adverse effects on the patient. Nanomedicines are produced, engineered, and industrialized at the cellular, chemical, and macromolecular levels. This study describes the various aspects of nanomedicine such as governing outlooks over high use of nanomedicine, regulatory advancements for nanomedicines, standards, and guidelines for nanomedicines as per Therapeutic Goods Administration (TGA). This review also focuses on the patents and clinical trials based on nanoformulation, along with nanomedicines utilization as drug therapy and their market value. The present study concludes that nanomedicines are of high importance in biomedical and pharmaceutical production and offer better therapeutic effects especially in the case of drugs that possess low aqueous solubility. The factual data presented in this study will assist the researchers and health care professionals in understanding the applications of nanomedicine for better diagnosis and effective treatment of a disease.

## INTRODUCTION

Over the past two decades, several studies explored and heralded nanotechnology as a "modern scientific breakthrough". The field allows for interconnected platforms based on solutions to unmet needs and problems in a variety of areas such as physics, chemistry, biotechnology, engineering, and especially in medical sciences; providing a range of possibilities in a variety of traditional research areas specifically in areas of medical sciences (*Navalakhe & Nandedkar, 2004*). Nanotechnology is gaining specific popularity in many of

**Table 1  List of marketed nanoformulations.**

| Formulation | Commercial name | Active ingredients | Indications |
|---|---|---|---|
| Liposomes | Ambisome® Visudyme® Abelcet® | Amphotericin Verteporfin Morphine | Fungal infections Macular degeneration Pain reduction |
| Pegylated liposome | Caelyx® | Doxorubicin | Cancer |
| Pegylated Liposome | Cimza® Somavert® | Rh-a/b Fab Fragment against TNF-α | Rheumatoid arthritis |
| Protein drug conjugate | Kadcyla® Abraxane® | Anti-HER2-GI/DM1 Albumin bound paclitaxel | HER2-positive breast cancer |
| Nanocrystal | Lipidic® Rapammune® Emend® | Fenofibrate Sirolimus Aprepitant | Hypercholesterolameia Nausea |
| Nanosuspension | Risperadol® Consta® | Risperidone | Schizophrenia |
| Emulsions | Neoral Lipuro | Cyclosporine Propofol | Immunosuppression Anaesthesia |
| Vaccines | Pandemrix | Split vision | Immunization |
| Polymeric nanoparticles | Renagel® Copaxone® | Sevelamer Glatiramer acetate | Hyperphosphataemia RR-MS |

these fields as an important scientific method for addressing major problems such as enhanced and precise medicine, reduced adverse effects/no toxicity risks, and meeting previously unmet patient needs in a suitable manner (*Shrivastava & Dash, 2009*). Nanomedicines have been engineered, produced, and industrialized at the chemical, cellular, and macromolecular levels in recent decades. The area of nanomedicine, which includes nanopharmaceuticals, nanoimaging agents, and theranostics, has resulted in increased creativity and significant advancements in disease detection, tomography, prevention, and care (*Shrivastava & Dash, 2009*; *Sahoo, Parveen & Panda, 2007*).

The rapid development of novel nanomedicines has shown potential in terms of enhancing public health and quality of life. Nanomedicines' nanoscale size (1–100 nm) and large surface area offer appropriate platforms for accessing biological targets and interacting with cells and tissues in a highly precise manner (*Sahoo, Parveen & Panda, 2007*). Increased funding and the effective implementation of multidisciplinary technologies in academia and industry have resulted in more advanced, dependable, and novel nanomedicines like polymer-conjugates, nanoparticles (Lipidic and Polymeric both), and liposomes. A list of marketed nanoformulations is represented in (Table 1).

These may lead to a variety of strategies, which are primarily supported by: (a) the utility of new fabrication processes, materials, and techniques (*e.g.*, surface/chemical modifications to overwhelmed concerns related to the stability of formulation and/or targeting ligands to cellular membranes or even organelles present inside the membrane); (b) the formulation in nano-scale size (*Chithrani, Ghazani & Chan, 2006*) which allows it to surpass through some major physiological obstacles, resulting in potent drug targeting; (c) the entrapment of necessary volumes of medicaments of various nature while shielding them from hostile setting, and achieving availability of medicament at the

required site in required concentration without impacting the co-existing healthy tissue while improving their toxicity profile (*Gaspar et al., 2014*).

Though a significant demand for supporting these nanomedicines, their unique properties have presented significant obstacles to industry and regulatory agencies. Furthermore, there has been a shortage of novel design criteria, precise procedures, and groundbreaking methods to classify these nanomedicines at clinical, physicochemical, and biological stages, which could have contributed to the failure in late-stage trials in a variety of instances (*Soares et al., 2018*). Therefore, key issues have progressively challenged the regulatory system for these revolutionary nanomedicines, to create a break to clear the guidance associated with their development.

## Governing outlooks on enhancement of nanomedicine

In recent decades, there have been several licensed nanomedicines having multi-dimensional applications in the biomedical field, but the lack of novel design criteria, precise procedures, and groundbreaking methods related to production at preclinical stages and depiction of these nanomedicines have hindered their probability related to further advancement in clinical fields (*Allen & Cullis, 2013*). Despite the attempts that have already been made, global regulatory developments must be established. Even though several significant measures have already been implemented over the last 5 years, strong cooperation between regulatory authorities/bodies is still required. Instead, the methods used currently to produce medicinal products having traditional background have been altered to assess the compatibility and the safety profile of experimental nanomedicines (*Dorbeck-Jung & Chowdhury, 2011*). According to the regulator, the conditions associated with the API of nanodrugs should be specified within the regulatory structure. Biological entities are subject to the same restrictions like proteins, peptides, and antibodies that come with new chemical entities (NCEs) and potential therapeutic active drugs (*Gaspar, 2010*). In general, a constraint to the management of these nanomedicines is directly related to their special properties. The advancements discussed for lipid nanoparticles, cubosomes, and other polymeric structures (including micelles), indicate that the therapeutic use of these more advanced and complex nanomedicines is highly dependent on flexibility. However their properties are flexible not just through slight adjustments in their raw materials, but also through minor alterations in their manufacturing processes. As a result, these changes lead to minor structural modifications in the bioactivities and bio-distribution behaviors (*Gaspar & Ducan, 2009*).

Besides, investigators and formulation scientists regularly link medicines and drug candidates, molecules used for targeting, as well as elements usually used for tracking and imaging to nanotherapeutics. A requirement for new quality control assays and some robust methods are for the effective monitoring and evaluation not only their physicochemical characteristics, such as size and size variability, shape, and charge but also because of their size and physical characteristics, nanomedicines are well known for their capacity to interact with immune cells and adsorb plasma proteins (*Dobrovolskaia & McNeil, 2007*). As a consequence, biocompatibility and autoimmunity must be taken into account during the preclinical evaluation. The therapeutic efficacy and dosage regimen, as

well as the administration pathway and targeted disease setting, all lead to an accurate estimation of toxicity during development. When molecules are entrapped and delivered to their specific destinations around the globe, nanomedicines have shown the potency and lower toxicity profile of conventional drugs, and also their biocompatibility (*Soares et al., 2018*; *Allen & Cullis, 2013*). On the other hand, numerous predicted nanomedicine classes, like quantum dots, dendrimers, and carbon nanotubes, have gained recognition. However, owing to the potential toxicity profile and immunological negative impacts of these innovative drugs, the clinical aspect of these drugs could be negotiated for many years (*Caminade & Majoral, 2010*).

Adapting manufacturing methods has been a new hurdle in the research, development, and medicinal translation of these nanodrugs. Due to the complex properties and reactivities of modern drugs, major production-related issues face contemporary pharmaceutical science at manufacturing plants, presenting a threat to their scale-up capability. It is important to identify and monitor important elements during each manufacturing step. Using QbD *i.e.*, quality by design concept, involving processes such as PAT *i.e.*, process analytical technology (PAT), will maintain a quality measurement technique that is online/at-line (*Stanczyk, Lee & Santen, 2007*). Anticipating and comprehending some of the critical points of progress helps us in the usage of automated systems to solve hiccups when they emerge. These foresighted plans resulted in the acceptance and implementation of ICH guidelines Q8, Q9, and Q10 as regulations for new pharmaceutical manufacturing. Furthermore, the important consequences for current pharmaceutical products explored in this paper may pave the way for the development of new and highly efficient manufacturing processes for future nanomedicines (*Merchant et al., 2009*).

Companies and governmental agencies in countries like Japan, USA and Europe have traditionally worked to enact comprehensive regulatory frameworks *via* ICH. Nonetheless, different points of view continue to serve as the basis for many protocols implemented by the USFDA, and the EMA. A few standardized assays and methods are often needed to examine problems that have a substantial effect on the *in-vivo* safety/ efficacy of nanotherapeutics (*Bawa et al., 2008*). The core complexity, however, is associated with the development of receptive assays for recognizing low doses of nanocarriers, discriminating them from modified aggregates, or from metabolized forms. Surprisingly, alternative imaging technologies, such as techniques used for cellular imaging and fluorescence, are anticipated and researched as tools for overcoming the limitations associated with it and pursuing healthcare breakthroughs. Another impediment to the regulation of these nanoproducts is the need of providing data before and during the product's life cycle, which necessitates in-vivo animal trials and therapeutic procedures (*Sainz et al., 2015*). The European regulatory regime allows for "scientific counseling services" from regulators to applicants for the benefit of Authorisation Applications (MAA), making for a smooth translational into clinical applications from the early stages of R&D. By that the impact of major avoidable obstacles in the process, could contribute to more harmonized development, driven by tremendous advancements in advanced techniques of nanomedicine. This can rapidly contribute to more organized progress,

driven by significant developments in advanced pharmaceutical strategies that can help to minimize the impact of numerous roadblocks encountered during the production process (*Mitchell et al., 2021*). Currently, the *European Medicines Agency (2015)* (EMA) has established a working group for the scientific evaluation and control of medicines, with a particular emphasis on concerns concerning the safety, feasibility, and efficacy of nano-products. Furthermore, this organization has prepared papers known as "orientation notes" that address important problems that candidates in the field of nanomedicines must remember (*Duncan & Izzo, 2005*). Despite the absence of detailed nanomedicine protocols, regulatory regimes in the United States (FDA), Europe (EMA), and Japan (PDMA/MHLW) have coordinated since 2009–2010 to gain large perspectives in the field of novel nanomedicine production (*Bawa et al., 2008*). Big pharmaceutical companies are now displaying a great deal of interest in the clinical development and "proof of concept" of these nano-systems. All these considerations would help to improve protocols for evaluating drug efficacy, targeting ability, toxicity control, and the safety of novel nanomedicine medicines (*Ventola, 2012*; *Oberdorster, 2009*). Furthermore, pharmaco-economic studies will be needed for the demonstration of the social and economic added value of these nanomedicines over established treatments prior to commercial exploitation, and critical aspects such as a rise in quality-adjusted life expectancy years *i.e.*, QALEYs or cost related to hospitalizations soon would also be included in the advancement of these theories (*Nadeem, 2020*).

## Regulatory advancements for nanomedicines of the "next generation"

The regulatory environment surrounding the production and evaluation of nanomedicines has been under increased scrutiny since the organization of the International Conference on Harmonization (ICH) in the early nineties. The predicted development necessitates more and more evidence of dominance in therapeutic effectiveness from new and emerging novel methods, as well as intensified pressure for pharmaco-economic evaluation (*Ehmann et al., 2013*). The historic evidence of non-inferiority is no longer applicable in certain treatment contexts, but clinical recommendations for the vast majority of applications are continually reviewed and modified. Furthermore, new opportunities in healthcare administration, such as health technology assessment (HTA), emphasize the importance of incorporating a variety of treatment methods that are complementary to the therapeutic classical oncology paradigm (*Patra et al., 2018*). Biopharmaceutical advances in several clinical areas have often shown their inherent superiority, outperforming other nanotechnology-based approaches. It is important to remember that the bulk of the modern structures are based on substances that were new in the late 1970s as well as early 1980s but are now no longer usable, including doxorubicin and others (*Gaynes, 2017*). Nanosimilars, which are the combination of generic drugs and nanocarriers used as an excipient (novel) are the major sticking point in the legislative discussions. Furthermore, the discussion involving formulations that are similar and are within the non-biological complex drugs (NBCDs), revealed a slew of critical issues in specific formulations (iron oxide nanoparticles, liposomes, polymeric micelles) (*Raj et al., 2019*). Both aspects are also subject to increase congressional oversight.

Simultaneously, the EMA's ITF (innovation task force) had begun to gather data, and in 2009, the adhoc expert group in Nanomedicines initiated a series of activities, including the first global conference on the topic, which put together regulators and shareholders from the United States and Japan. During the same period, the European Union dispute on nanomaterials classification related to the "similarity" topic spurred the formation of a TI Pharma (Top Institute Pharma)-based international expert group in the Netherlands (NBCDs expert group) comprising of a variety of experts from academia and industry (*Kolosnjaj, Szwarc & Moussa, 2007*). The FDA, PDMA/MHL, Canada, Switzerland, Australia, and others all initiated their programs, some of which were coordinated with the EMA. The EMA issued a series of guidance documents on nanostructured materials (Reflection Paper for Generic Nanoparticle on Non-Clinical Studies Iron Medicinal Product Applications e Ema/Chmp/Swp/100094/2011), liposomal comparable models (Reflection Paper for Intravenous Liposomal Products Developed with Liposomes Ema/Chmp/Swp/100094/2011 on the Data Requirements). Recently, a list of the concepts guiding such documents was published (*European Medicines Agency, 2015*).

## Standards and guidelines for nanomedicines as per TGA

There are several nanomedicines on the market. Still, there is no standard methodology for studying the safety and effectiveness of these nanomedicines, so their development is impeded. The API for nanomedicines, from a regulatory standpoint, requires that the specifications be evaluated within the regulatory framework. The regulatory requirements must be followed by the functional entities present in nanomedicine. The regulatory criteria for medicinal plants and chemical entities have been established (*Tambe et al., 2019*). The new pharmaceutical development rules described in ICH recommendations Q8, Q9, and Q10 aid in the monitoring and management of key process parameters. The different regulatory authorities engaged in the regulation of nanomedicines include the USFDA, TGA, and EMA (*Fan et al., 2012*).

## TGA

TGA refers to a broad variety of techniques used in the fabrication and design of methods and processes by regulating size and form at the nanoscale. The TGA regulates goods more effectively since it has a high degree of competence in evaluating new technology. Furthermore, it has the legal power to request extra evidence in support of the safety evaluation of novel materials, and it generally works with applicants who have the technical competence to properly address major safety concerns (*Faunce, 2007*). TGA offers development programs in which the physicochemical characteristics of nanomedicines are compared to those of traditional medications. The therapeutic advantages of the produced nanomedicines can also be assessed. It involves investigating the kinetics of nanoparticles as well as the *in-vitro* activity of nanoparticles. The toxicity of nanoparticles *in vivo* and *in vitro* is next evaluated, followed by risk management and risk reduction. As a result, the TGA itself is on the frontline of regulating nanomedicines (*Faunce, 2009*). Wherever feasible, the Therapeutic Goods Administration (TGA) aligns the therapeutic drug legislative mechanisms with those of similar

international regulatory counterparts. Technical data standards are loosely matched with those outlined in applicable EU and ICH guidance. As EU and ICH technological recommendations are incorporated into Australian law, they provide sponsors with instructions to help them fulfill regulatory criteria (*Collins & Varmus, 2015*).

The legislative mechanisms in Australia are largely well adapted to the role of overseeing technology. Nanotechnology-based materials are subject to all regulatory mechanisms. There was no urgent need for significant improvements to the regulatory regimes, although small modifications were needed. The TGA is well-positioned to oversee goods containing nanomedicines because it primarily works in data-rich ecosystems and has a strong degree of experience to put to bear on the evaluation of emerging technology. TGA has the legal right to request supplementary evidence in favor of current content protection assessments (*Muthu & Feng, 2009*).

## Patented nanoformulation

In order to enhance the commercial utility of nanomedicines, several inventors have patented their nanoformulations. Table 2 presents some of the patents based on nanoformulations.

## Clinical trials on nanomedicine

Until 2015, the USFDA licensed 13 nanomedicines to treat various types of pathology. Recently, there has been a surge in the commercialization of these products, as well as a large rise in clinical trials. According to existing statistics, there are 104 ongoing clinical trials, with 82 (78.8 percent) in oncology and 94.8 percent in step I, II, or I–II. 14.6 percent in the EU, 12.2 percent in China, and 4.9 percent in other countries (*Albercht et al., 2018*). *James, Betty & Warren (2010)* studied BXQ-350, a nanomedicine agent, in advanced solid tumours, including malignant brain tumours. As BXQ-350 interacts with Saponin C, a naturally expressed protein in humans, it produces DOPS (are fat molecules of Nanobubbles). This formulation has the potential to selectively target and destroy cancerous tumour cells. The results of Phase 1 clinical trial reveal that the patient tolerates the agent well, that there is no dose limiting toxicity, and that there are no significant adverse reactions in the treatment. This study would have 40 new patients from around the country as part of an extended step IB trial (*Alvarez, Aranega & Molina, 2018*).

Clinical trials of anti-cancer nanomedicine were classified into four categories: liposomes, polymeric conjugates, polymeric nanoparticles, micelles, and so on. Approved nanomedicine for cancer has been developed to use the EPR impact principle, with a group of nanomedicines achieving improvements in nanomedicine behaviour by ligand-controlled targeting. EPR-based drugs aim to increase effectiveness and tolerability by modifying medication pharmacokinetics and biodistribution (*Hare et al., 2017*). According to *Lammers et al. (2020)*, clinical translation for cancer nanomedicine is limited due to a variety of factors such as a lack of understanding of biological barriers in the body, misunderstanding of concepts used in drug delivery, cost-effectiveness, scale up and manufacturing process, and regulatory issues. The analysis of pre-clinical studies over the last 10 years shows that only 0.7 percent of the dose has been delivered to tumours (*van der Meel, Lammers & Hennink, 2017*).

**Table 2 Patent based on nanoformulation.**

| S. no. | Inventor name | Work done |
| --- | --- | --- |
| 1 | James Q. Lillard, Jr. Rajesh Singh | The investigators presented nanoparticles formulation and methods for supplying a bioactive agent to a plant. Work highlightd that nanoparticle structure contains a coronatine-coated nanoparticle that is intended to deliver one or more bioactive agents *via* plant stomata. Nanoparticles may include one or more bactericides, molluscicides, miticides, fungicides, nemanticides, insecticides, herbicides, acaricides, plant nutrients, fertilizers, growth regulators, or combinations thereof (*Lillard, Singh & Singh, 2015*). |
| 2 | Shaker A Mousa | Wok involves loading of nanoformulation comprising nanoparticles. In each nanoparticle, a transformed chitosan polymer encapsulated minimum one vitamin D analog, at least one vitamin D metabolite, or combinations alike. The changed chitosan polymer contains chitosan that has been covalently linked to at least one person was selected from the fatty and variations respectively. The structure was made up of a solvent and a nanoformulation, with nanoparticles dispersed in the medium (*Wu et al., 2021*). |
| 3 | Anamaria Ioana Orza | The invention relates to new anti-wrinkle and anti-aging nanoformulations, composed of non-toxic mesoporous silica nanoparticles, natural plant extracts (as the case may be: pomegranate oil, fennel oil, rosemary oil, chamomile oil, jojoba oil, rosehip oil), biologically active agents (acetyl hexapeptide-8, aspartic acid), vitamins and others (*Orza, 2017*). |
| 4 | Vuong Trieu | The present invention relates to therapeutic agent nanoparticles, nanoparticle formulations appropriate for injection, means for administering therapeutic agents and for treating disorders and conditions treatable by the therapeutic agents using the formulations, in particular, the formulation and characterization of nanoparticles containing taxanes such as paclitaxel (*Desai et al., 2009*). |
| 5 | Vuong Trieu | The present innovation relates to methods for guiding the engineering of nanoparticle drugs for intravenous administration using pharmacokinetic criteria and other studies. The methods of the present innovation are particularly useful in the preparation of nanoparticles containing cytotoxic drugs for cancer care (*Trie, D'Cruz & Desai, 2012*). |
| 6 | Chiara Nardon Dolores Fregona Leonardo Brustolin Nicolò Pettenuzzo | The present development relates to mononuclear coordination composites of Au and Cu, pharmaceutical nanoformulations based thereof, the relative method of synthesis and encapsulation of the compounds in macromolecules, supramolecular aggregates, or nanostructures, as well as their use for the diagnosis and/or treatment of neoplasia (*Chiara, Leonardo & Nicolo' Pettenuzzo, 2016*). |
| 7 | Liangfang Zhang Soracha Kun Thamphiwatana Che-Ming Jack Hu Ronnie H. Fang Brian T. Luk | The present innovation describes processes, formulations, and pharmaceutical formulations for preventing and/or treating infection in a target by a platelet-targeting microbe, utilizing, among other things, an appropriate volume of a nanoparticle composed of an inner center, a non-cellular substance, and an outer surface composed of a cellular membrane extracted from a platelet (*Zang et al., 2018*). |
| 8 | Liangfang Zhang Zhiqing Pang Ronnie H. Fang Che-Ming Jack Hu | The current invention pertains to toxin therapy in a patient. The present invention describes compositions for reducing or neutralizing the influence of a toxin in a topic by employing, among other things, an efficient volume of a nanoparticle with an inner center composed of a non-cellular substance and an outer surface composed of a cellular membrane obtained from a source cell. Toxins such as organophosphate overdose are examples of acetylcholinesterase (AChE) inhibitors (*Zhang et al., 2018*). |
| 9 | Rambhau Devraj, Pranati Chhatoi, Naga Hemanth Kumar Parvathabhatla, Anand Vasant Deshmukh, Krishna Kaushik Chintabhatla | The present disclosure is geared against a secure nanodispersion comprised of an aqueous dispersion medium, a dispersed process, a surface-active agent, and alternatively, an additive, whereby the aqueous dispersion medium includes a nanodispersion stabilizing vehicle base portion that enhances the nanodispersion's long-term physiochemical properties (*Devraj et al., 2015*). |
| 10 | Hemanta Koley, Priyadarshini Mukherjee, Dhrubajyoti Nag, Ritam Sinha, Manoj Kumar Chakrabarti | An OmpA formulation consisting primarily of (a) OmpA protein as active molecule obtained as a product of the OmpA gene inserted in a plasmid containing a novel set of forward and reverse primers and (b) Alginate chitosan nanoparticles as a vehicle (*Koley et al., 2019*). |

| S. no. | Inventor name | Work done |
|---|---|---|
| | **Table 2 (continued)** | |
| 11 | Shaker A. Mousa | Each nanoparticle is encapsulated in a shell that contains sulfated non-anticoagulant heparin (SNACH), which can be used with or even without hydrophobic anti-angiogenesis Tyrosine Kinase inhibitors. The SNACH is ionically or covalently bound to the shell. In the shell, a polymer from the poly (lactic-co-glycolic acid) (PLGA), chitosan, chitosan-alginate, and NIPAAM-APMAH-AA group is included (*Mousa, 2017a*). |
| 12 | Minoru Toriumi Toshiro Itani | The current innovation describes a tool for planning a composite near-infrared lasernano formulations triggered by the near-infrared laser irradiation nanoformulation 980 nm, whereby the object transmitting means converting the light-triggered photosensitive nanoparticles release pharmaceutical micelles, to diagnosis and treatment of visible light through diagnosis and treatment of secondary medication problem (*Toriumi & Itani, 2018*). |
| 13 | Jiang Hulin<br>Xing Lei<br>He Yujing<br>Cui Pengfei<br>Huang Wei | The proposed design discloses a type of functional albumin nanoparticles formulation and preparation method; functionalized nano-albumin formulation of functional albumin, a metal ion, and pharmaceutical composition; functional metal ions simultaneously form a coordinate bond with albumin and pharmaceutical, inducing self-assembly of nanoparticles (*Jiang, Xing & Yang, 2019*). |
| 14 | Nan Kaihui | The innovation relates to a method of preparing a Legumain sensitive nano preparation for multi-step release of adriamycin amycin/slow release of curcumin, as well as an operation. A Legumain sensitive nanogel is prepared based on the biological feature that tumour tissues transmit Legumain abundantly, resulting in a double targeting impact (active and passive targeting) of the tumour tissues and a reduction in secondary damage on normal tissues caused by adriamycin amycin (*Wu et al., 2021*). |
| 15 | Nygaard Halvor Langmyhr Eyolf | The present discovery discloses a scientific area Scotogramma belonging to agricultural chemicals prepared armyworm decoy compositions and formulations become prepared nanoparticles. The amphiphilic copolymer is shaped by self-assembly in an aqueous solution with a distinct core-shell framework nano-sized micelles, developed by electrospinning technique nanometers pesticides (*Halvor & Eyolf, 2010*). |
| 16 | Nobuhiko Yui Tooru Ooya Ikuo Sato | The present discovery is about a procedure and implementation of a modified hyaluronic acid preparation of single walled carbon nanotube diagnosis and treatment of drug nanoparticles sensitivity reducing agent, which can effectively solve the low targeting anti-cancer drug doxorubicin in tumour (*Yui, Ooya & Sato, 2004*). |
| 17 | Martin Hurby Ingrid Brezaniova Vladimir Kral | The invention is directed to photoactivatable nanoparticles for photodynamic applications that include a photosensitizer, a (C8 to C22) fatty alcohol, and a polymeric surfactant, preferably poly(ethylene oxide) monomethyl ether-block-poly(eta-caprolactone), with the size of the nanoparticle ranging from 1 to 1,000 nm and the core solid at 4 °C. The present discovery often applies to a means of producing the photoactivatable nanoparticle, as well as a medicinal composition comprising it and its use (*Hurby, Brezaniova & Kral, 2019*). |
| 18 | Cheng Zhang Martin Fransson Bing Zhou | The invention aims to include a nano-carbon preparation for labeling as well as a tool for making it. The invention's nano-carbon preparation is increased in dispersibility and consistency, has good bio-safety, is easy to prepare, has a low manufacturing cost, and can be used to trace tumour cells in the lymphatic system and mark lesions or biopsy sites in the digestive tract (*Zhang, Fransson & Bin, 2006*). |
| 19 | Shaker A Mousa | This disclosure relates to novel formulations and Nanoformulations as described in the specification, as well as formulas containing a mixture of HCV protease and polymerase inhibitors, with or without interferon, as well as anti-fibrotic/anti-hemolytic agents composed of naturally derived polyphenols/thiols and non-anticoagulant GAGs. These compounds are potent antiviral agents, particularly in inhibiting the function of various genotypes of Hepatitis C virus (HCV) (*Mousa, 2017b*). |

(Continued)
| S. no. | Inventor name | Work done |
|---|---|---|
| | **Table 2 (continued)** | |
| 20 | Ana Brotons Canto<br>Carlos Gamazo De La Rasilla | The proposed model discloses nano-formulations for treating irritable bowel syndrome, as well as by the zein nanoparticles coated on the outer zein nanoparticles consisting of *E. coli* outer membrane vesicles; the zein nanoparticles coated with antibiotics, antibiotics are antimicrobial peptides; antimicrobial peptide sequence as a RIVVIRVA, KIWVIRWR, RIWVIRWR, RIWVIWRR, IWVIRWR, RWVIWRR, VVIRVA, WVIRWR or in WVIWRR (*Canto et al., 2018*). |
| 21 | Shaker A. Mousa | Nanoparticles make up a nanoformulation. Each nanoparticle is encased in a glycosaminoglycan shell (GAG). The GAG is bound to the shell either ionically or covalently. The GAG is chosen from among sulfated non-anticoagulant heparin (SNACH), super-sulfated non-anticoagulant heparin (S-SNACH), and a combination of the two. The shell is made of poly (lactic-co-glycolic acid) (PLGA), polyethylene glycol (PEG)-PLGA, PLGA-Polycaprolate, or calcium alginate. A technique of utilizing the nanoformulation to manage cancer in a subject involves prescribing a therapeutically effective volume of the nanoformulation to the recipient for cancer treatment (*Mousa, 2018*). |
| 22 | Shaker A Mousa, Mohammed H Qari,<br>Mohammed S. Ardawi | A nano-composition and a process for using it. The composition includes nanoparticles. Each nanoparticle has a shell that contains lycopene. The shell is made up of electrostatically bound oligomerized (−)-epigallocatechin-3-O-gallate (OEGCG) to chitosan. The implementation strategy for the formulation entails delivering the nano-composition to an individual (*Mousa, Qari & Ardawi, 2017*). |
| 23 | Omid C Farokhzad<br>Pedro M Valencia<br>Xiaoyang Xu<br>Xueging Zhang<br>Mingming Ma<br>Nazila Kamaly | The four major components of sub-100 micron multimodal nanoparticles are: (1) A target factor that can attach to cells, tissues, or organs of the body selectively; (2) A diagnostic agent, such as a fluorophore or NMR contrast agent, that allows nanoparticles to be visualized at the site of delivery (3) An outer "stealth" coating that enables the particles to avoid detection by immune system components; and (4) a biodegradable polymeric substance that forms an inner center capable of carrying therapeutics and releasing payloads at a sustained pace during systemic, intraperitoneal, or mucosal administration (*Farokhzad et al., 2013*). |
| 24 | Shaker A. Mousa | A nanoparticle-containing nano-composition, a method of forming the nano-composition, and a method of using the composition are all described. Chitosan, poly L-Lysine, poly L-Arginine, methylated chitosan, are polycationic polymers that are ionically attached to one or more polyanionic Glycosaminoglycans (GAGs) (*Davis, Davis & Mousa, 2010*). |
| 25 | Wang Xuemei<br>Zhao Chunqiu<br>Wang Jianling<br>Ren Fa | The present discovery discloses a nanocomposite porphyrin derivative of nanoformulation for use in photodynamic therapy and photothermal therapy for rheumatoid arthritis, as well as it's use (*Wang et al., 2017a*). |

*Bobo et al. (2016)* discovered that since the mid-1990s, a total of 13 nanomedicines have been licensed for novel therapeutic indications per 5 years. This involves the approval of experimental materials as well as their use for additional therapeutic indications (for example, Abraxane® has been accepted for a variety of indications) (*Bobo et al., 2016*). Four nanomedicines are produced by Katoka that are being run for clinical trials by Chiba, Japan-based nanomedicine manufacturer, Nano Carrier and Nippon Kayaku, a Tokyo-based global chemical and drug firm. One of the above drugs is in phase 3 of clinical trials for pancreatic cancer, and the drug release is caused by the acidic condition of the tumour (*Bourzac, 2016*). The mice of various sizes were used, and a dosage of various shaped calcium phosphate nanoparticles was administered. Pancreatic tumours and their vasculature were imaged every 10 min for the next 10 h. The substance was not inserted

into these nanoparticles, but they can be used in live animal microscopy. *Koley et al. (2019)* discovered the particles which are as small as 30 nanometers which leak into pancreatic tumours, but larger particles occur in blood at random due to nanoeruptions.

Doxil and Abraxane are first-generation nanoparticle therapeutics that have already gained acceptance in the clinical cancer research world. Nanoparticle-based therapeutics can sustain their potential of showing increased medication concentration in tumours with fewer side effects. Most of the nanomedicine is focused on passive targeting that takes advantage of the EPR effect; thus, active targeting technology is being created (*Wang, Billone & Mullett, 2013*). Polymer drug conjugates, micelles, protein-based carriers, liposomes, polymeric nanoparticles, and inorganic nanoparticles are the most common types of nanoparticles accepted in the final stage of clinical trials.

Within the tumour microenvironment, nanoparticles can be engineered to deliver medications sequentially and at specified molar ratios, allowing for maximum synergy that is not achievable with traditional drug delivery methods. For instance, Vyxeos, a liposomal formulation used in the treatment of acute myeloid leukaemia (AML), daunorubicin and codelivers cytarabine in a set molar ratio of 5:1. When compared with a standard daunorubicin and cytarabine regimen, Vyxeos showed enhanced efficacy in two phase 2 clinical trials. Vyxeos was authorised by the FDA in August 2017 for the treatment of AML based on results from five clinical trials, including a pivotal phase 3 trial that met its primary endpoint (*Ventola, 2017*).

The journey from original discovery to the marketplace for a novel therapy takes about 10 years in a clinical trial. Clinical trials take an average of 6 to 7 years to complete. Phase 1 and 2 takes approx. 1 year to 2 years to complete while Phase 3 lasts from 1 to 4 years. Following Phase 3, a pharmaceutical company may submit a New Drug Application (NDA) or a biologics licence application (BLA) to the Food and Drug Administration for approval of the treatment (FDA). The FDA next evaluates the results of all stages of the study to determine whether the drug will be approved and the pharmaceutical company will be able to begin marketing it to the general public.

"Post-Approval Research and Monitoring" is a term used to describe Phase 4. Long-term negative effects may take time to manifest, making this a crucial stage (https://www.antidote.me/blog/how-long-do-clinical-trial-phases-take).

## NANOMEDICINES AS DRUG THERAPY

### Cancer

Over conventional medication construction and distribution, nanotechnology can transform pharmacokinetics and distribution, enhancing potency and reducing side effects (*National Cancer Institute, 2009*; *National Cancer Institute Alliance for Nanotechnology in Cancer, 2015*). Drug portability, agility, and detainment, as well as selective containment in tumour tissue, are improving, and vectors with potentially successful distribution are being created, aided by the size and surface properties of nanoparticles (*Bhandare & Narayana, 2014*).

### Passive targeting

Because of their small size and surface properties, nanoparticles can pass through blood vessel walls and into tissues. Since tumours have leaky blood vessels and poor lymphatic drainage, nanoparticles may accumulate in them (*Sebastian, 2017*). EPR is the name given to this passive targeting impact. The EPR improves medication distribution in solid malignant diseases such as breast cancer (*Wu et al., 2017*). However, passive targeting cannot eradicate the possibility of nanocarriers accumulating in tissues with perforated blood vessels, such as the liver or spleen (*Tran et al., 2017*).

### Active targeting

Active targeting is focused on the targeted targeting of nanoparticles and the molecules expressed on the surfaces of cancerous cells. The theory is used in this targeting to distribute medicines by causing the cell to ingest the nano-carrier into the cancerous cell (*Sebastian, 2017*). When active targeting is paired with passive targeting, the association of carried drugs with healthy tissue is reduced much more. *Via* nanotechnology, low drug doses will contribute to greater tumour reduction and increased chemotherapeutic efficacy. Targeting moieties, such as monoclonal antibodies or receptor ligands, are bound to nanocarriers with high affinities for these targets, allowing the nanocarriers to associate and accumulate there (*Soares et al., 2018*). Conjugation of these ligands has the potential to reduce non-specific uptake of nanocarriers to tissue other than tumour tissue (*Wu et al., 2017*).

## Malaria

The nanomedicine used for targeting contaminated RBCs and hepatocytes occasionally achieves antimalarial agent targeting. There are two approaches: proactive and aggressive targeting. Passive targeting involves the alteration of the surface with hydrophilic polymers, while active targeting involves the attachment of various ligands to the nano-platforms (*Garg et al., 2017*).

### Passive targeting

The primary goal of targeting is to increase a drug's plasma maximal concentration (Cmax), which is proportional to its toxicity, and effectiveness is proportional to the region under the curve (AUC) of drug plasma concentration. A higher drug concentration increases drug contact with compromised RBCs and parasite membranes (*Gogoi, 2017*).

### Passive targeting using traditional nano-platform-based DDS

The intravenous path is used less in malaria care than in leishmaniasis therapy due to the differences in the host cell which is infected. RBCs are neither phagocytically nor endocytically involved. It should be remembered, however, that the nanocarriers that are parentally administered are quickly picked up and transports the drug inside the macrophages by the MPS cells (*Pam et al., 2017*).

### DDS with a hydrophilic surface-modified nano-platform based on passive targeting

The surface alteration of nanoplatforms can be performed by using hydrophilic polymers such as PEG (Poly-Ethylene Glycol), PAA (Poly-Acryl Amide), PVA (Poly-Vinyl Alcohol), PVP (Poly-Vinyl Pyrrolidine) and Polyxamer. It delays phagocytosis, increases the half-life of the medication in the serum, and allows for the regulation of the drug's biodistribution and pharmacokinetic profile. As a result of the hydrophilicity of the polymer, the polymer occupies the surface of the nanoplatforms and thereby preventing the blood plasma opsonins from accessing and attaching to the surface of nano carriers (*Müller & Wallis, 1993*).

### Active targeting

Active targeting of drugs associated with nanocarriers is accomplished by conjugating a cell-specific ligand at the carrier's surface, causing the medication to accumulate preferentially in the target cell or tissue (*Pam et al., 2017*).

### Hepatocyte targeting

The liver is suitable for nanoparticle absorption and targeting due to the open and relatively elevated perforations of the endothelial coating of sinusoids, which lead to the extravasation of larger particles (*Gogoi, 2017*).

### Erythrocyte targeting

*I. vitro*, nanoplatforms functionalized antibodies demonstrated full intolerance for pRBCs as opposed to non-infected erythrocytes. Glycosaminoglycans (GAGs) are negatively charged membrane molecules that attach to pRBCs (*Wu et al., 2017*).

### Targeting to brain

The parasite *P. falciparum* migrates to the brain in cases of cerebral malaria (CM). The nanoparticles improve drug permeability through a variety of pathways, including transcytosis or endocytosis, inhibition of the transmembrane efflux system (P-glycoprotein), opening tight junctions between endothelial cells, fluidization of the membrane, and toxic effects on the barrier, allowing for restricted permeabilization of brain endothelial cells (*Desai et al., 1996*). Endothelial cell endocytosis of nanoparticles was improved after drug administration into the brain. Strong lipid nanoparticles (SLNs) of the antimalarial medication quinine were prepared and conjugated with transferrin receptors for active brain targeting (*Gogoi, 2017*). The parasite *P. falciparum* migrates to the brain in cases of cerebral malaria (CM). The nanoparticles improve drug permeability through various mechanisms such as transcytosis or endocytosis, inhibition of the transmembrane efflux system (P-glycoprotein), opening tight junctions between endothelial cells, fluidization of the membrane, and toxic effects on the barrier, allowing for restricted permeabilization of brain endothelial cells. Endothelial cell endocytosis of nanoparticles was improved after drug administration into the brain. Strong lipid

nanoparticles (SLNs) of the antimalarial medication quinine were prepared and conjugated with transferrin receptors for active brain targeting (*Garg et al., 2017*).

## HIV

Drug delivery to the brain has been signifying based on the surface receptors but the volume of distribution was limited. Magnetic medication nanocarriers are magnetically applied to the brain for the treatment of HIV/AIDS, which is a non-invasive process (*Pam et al., 2017*). Under the influence of a static magnetic field, magnetic nanoparticles loaded with 3-Azido-2,3-dideoxythymidine-5-triphosphate (AZTTP) successfully cross the BBB, and the virus is released by systemic drug release, accompanied by a virus killing to suppress virus levels. *Chiappetta et al. (2010)* created efavirenz-loaded polymeric nanoparticles for the treatment of HIV in infants. This pharmacotherapy improves the bioavailability of efavirenz (20 mg/ml).

Various nanomedicine methods for antiretroviral medication oral distribution are being investigated and tested. The current solution is solid medication nanoparticle formulations, which are used to improve the bioavailability of poorly water-soluble products. Macrophages are also a major focus for HIV. Nanocarrier devices that exploit the mannose receptor have been suggested for HIV therapy (*Kaushik, Jayant & Nair, 2018*).

The application of nanotechnology to antiretroviral drug delivery holds promise for HIV cure because it has the potential to change tissue dissemination by targeting drugs to HIV reservoirs and increasing drug half-lives (*Curley, Liptrott & Owen, 2018*).

Targeted transmission to HIV reservoir sites will be extremely beneficial since many antiretroviral drugs do not reach these sites optimally, contributing not just to viral survival but also drug resistance growth (*Kumar et al., 2015*).

### Tuberculosis

The rich layer of mycolic acid in *M. tuberculosis* cell wall makes the distribution of antitubercular drugs (ATDs) difficult. Since antitubercular medications are hydrophilic in nature, they can cause adverse effects when used for an extended period of time. There are prerequisites for a carrier device that can increase antitubercular medication permeability (*Kaushik, Jayant & Nair, 2018*). This would results in lower levels of the medication being effective and fewer side effects. The creation of novel nanomaterial-based methods for the bioavailability and avoiding drug resistance development for enhancing site-specific targeting of antitubercular drugs.

When compared to its free medication, gentamycin encapsulation into liposomes decreased myocardial infection. The antitubercular drugs' antimycobacterial efficacy was improved after encapsulation in liposomes (*Lenjisa, Woldu & Satessa, 2014*).

Nanoparticles are more easily absorbed by cells than larger molecules, making them a promising transport and distribution mechanism. Because of the durability and continuous release of medications from nanoparticles, oral administration is feasible (*Kumar, Das & Patra, 2017*).

The use of various nanotechnology-based drug delivery systems, such as polymeric nanoparticles, strong lipid nanoparticles, liquid crystal systems, liposomes,

microemulsions, nano micelles, and metal-based nanoparticles (gold nanoparticles, silver nanoparticles, iron oxide nanoparticles) (*Nasiruddin, Neyaz & Das, 2017*), are an intriguing technique for improving on the most attractive properties of a formulation. Furthermore, particles reflect a future in which activity is guaranteed and complications correlated with TB care can be resolved (*Knox et al., 2019*).

This may increase bioavailability and, as a result, dosing duration, leading to improved patient compliance and drug effectiveness (*Da silva et al., 2016*).

## Market value of nanomedicine

Nanomedicines have been regarded as an analytical and innovative approach to propose drugs since the early days of nano-based technology, with a strong potential for the advancement of therapies for disorders with a poor prognosis, such as cancer, respiratory disease, viruses, neurodegenerative diseases, and non-curable diseases (*Nasiruddin, Neyaz & Das, 2017*). Although they have been widely used as drug delivery vehicles for targeting purposes since their invention, several nano-based materials are now being produced as therapeutic agents on their own. Among others are reactive oxygen species (ROS) promoters (*Costa et al., 2016*), hyperthermia agents (*Sims et al., 2017*), and X-ray enhancers (*Thiesen & Jordan, 2008*). Consequently, a growing number of large and small pharmaceutical firms are investigating the potential applications of Nanomedicine (*Pottier, Borghi & Levy, 2015*; *Leem, 2013*; *Ventola, 2012*). On the other side, as compared to the immense and fascinating research activities in both academia and clinics (*Gaspar & Ducan, 2009*; *Wang et al., 2017a*; *Tinkle et al., 2014*; *Weissig, Pettinger & Murdock, 2014*), the selection of marketed nanomedicines (among which the most popular Onivyde® (2015), Abraxane® (2005), Doxil® (1995), AmBisome® (1990)) remains inconsistently poor.

## Global nanomedicine market-forecast, 2017–2023

Nanomedicines have a strong ability to modernize the current scenario of disease prevention, care, and diagnostic processes. From 2017 to 2023, the global nanomedicine market (Global Opportunity Analysis and Industry Forecast from 2017 to 2023) is projected to rise (Fig. 1) at a compound annual growth rate (CAGR) of 12.6 percent, reaching $528 billion in 2019 and $2,611 billion in 2023 (*Ragelle et al., 2017*; *Business Wire, 2016*).

The global nanomedicine industry is being propelled by new technology for drug distribution, the benefits of nanomedicine in numerous public healthcare applications, an increase in government interest and investment, and the need for secure and cost-effective therapies (*Ragelle et al., 2017*). However, environmental concerns (risks associated with nanomedicine) and a lengthy approval procedure limit business expansion. Furthermore, the increased out-licensing of nano drugs and the expansion of healthcare facilities in developing economies are projected to provide numerous opportunities for business growth (*Business Wire, 2016*).

On a global scale, the nanomedicine market is segmented by use, indication, modality, and area. Drug distribution, vaccinations, regenerative medicine, medical testing, implants,

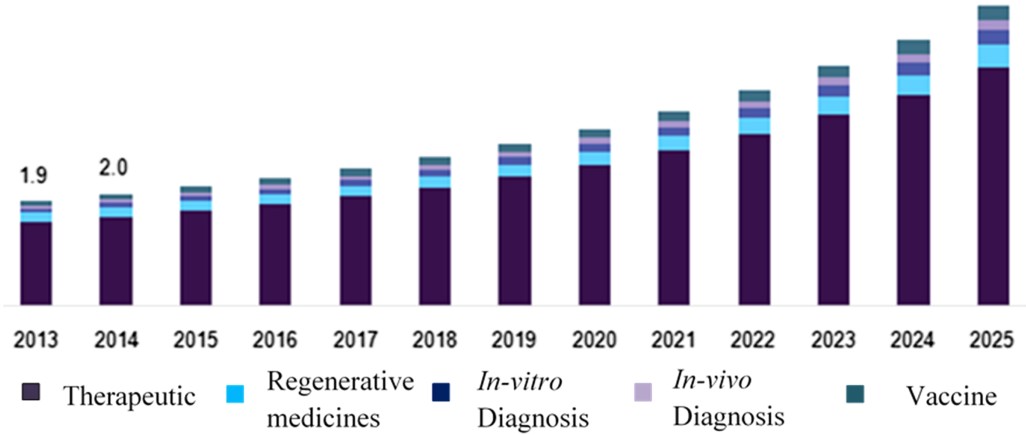

**Figure 1 Demand for nanomedicine in the United States by application, 2013–2025 (USD Billion).**

among others are application-based segments. Oncological diseases, immunological diseases, respiratory diseases, infectious diseases, psychiatric diseases, urological diseases, ophthalmological diseases, orthopedic disorders, among others are categorized by indication (*Pottier, Borghi & Levy, 2015*). The modality-based section, on the other side, is divided into diagnosis and therapies.

## Growing application of nanomedicine noted in cardiovascular and neurology segments

Cardiovascular, neurology, oncology, anti-infective, and anti-inflammatory are the most significant application categories of nanomedicine. Oncology is expected to dominate the overall industry over the prediction timeframe, with a valuation of $65.5 billion by 2019. When opposed to traditional treatments, nanomedicine has shown many advantages in the management of tumors (*Ragelle et al., 2017*). Furthermore, as opposed to non-targeted drug therapy, targeted drug delivery has many benefits in terms of accuracy by improved permeation and retention (EPR). Overall, there has been a greater emphasis on numerous research programs including the usage of nanomedicine for cancer treatment.

However, in recent years, the emphasis on nanomedicines has shifted to their usage in cardiovascular and neurology. It is mostly due to the increased access to medical records and technical advances. A CAGR of nearly 16.0 percent over the projected era is expected to develop in the cardiovascular segment (*Gogoi, 2017*).

## Nanomedicine and Asia Pacific market

Though nano-based products will still be in their early stages of development and advancement in Asia Pacific's developing economies, the sector may over the forecast period is expected to expand. The APAC nanomedicine industry, led by India, China, and Indonesia, is projected to be propelled by increased investments in R&D as well as the introduction of new drugs/agents and therapies. A deep pipeline centered on different

therapeutic areas is also expected to drive the development of the Asia Pacific nanomedicine sector (https://www.reportbuyer.com/product/5139404).

## Presence of patented nanomedicine to drive Asia Pacific market

The Asia Pacific region is expected to expand significantly, owing to increased contributions from China, India, and Indonesia. Furthermore, investments in the field of research and development activities shoot up for the launch of new therapies and medicines are expected to intensify this region's growth in the coming years (*SBWire, 2018*).

## Insights into industry

In 2016, USD 138.8 billion was the estimated market of global nanomedicines. Technological advancements coupled with appropriate applications in early disease diagnosis, preventive intervention, and chronic and acute disease prophylaxis are expected to fuel market growth. Nanotechnology is the nanometric miniaturization of larger products and chemicals, which has significantly revolutionized medication administration and will continue to dominate technical adoption until 2025 (*SBWire, 2018*).

Expected advances in nanorobotics as a result of increased investment from government agencies are expected to bring potential to the industry. Nanorobotics research programs aiming to kill cancer cells without disrupting adjacent tissues are expected to accelerate development until 2025 (Fig. 1).

A potential pipeline of goods is expected to propel the industry with potential development avenues focused on nano molecules and related innovations.
The involvement of approximately 40% of medicines in phase II clinical research is likely to result in some key commercialization during the next decade, influencing sales generation over the forecast era. Because of the tailored therapy options for eradicating genetic disorders, this technology is a promising option for precision medicine (*Desai et al., 1996*).

## Insights into product

Because of the inclusion of many drugs in this segment, In 2016, therapeutics described a key portion of revenue. Therapeutic drugs and instruments, as well as medication delivery mechanisms, are used to cure a variety of diseases (*Costa et al., 2016*).

Nanoparticles are a potential technology for drug delivery to the target site of action due to their unique physicochemical properties. Furthermore, nanotechnology-enabled drug delivery systems have two major advantages:

1. Improved cellular penetration.
2. Toxicity associated with the drugs is reduced.

These benefits are expected to draw attention to nanotechnology for drug delivery as a product launch, affecting growth throughout the forecast period. Growth in this segment is expected to be driven by the production of novel delivery products using different forms of nanoparticles (*Transparency Market Research, 2018*).

### Insights into application

Oncology, respiratory disorders, cardiology, orthopaedics, among other applications are included in the segmentation. In 2016, oncology led with a revenue share of about 47 percent. A significant number of medications in the clinical phases of development for cancer treatment, as well as developments in the implementation of therapeutic and diagnostic particles and devices, are to blame for a larger proportion of cancer diseases (*Gogoi, 2017*). The tool could be used for both active and passive cancer targeting. Cardiology, on the other hand, is projected to develop the most, with a CAGR of around 11.7 percent over the forecast timeline, owing to the availability of opportunities such as demand for novel therapeutic nanovectors, nanostructured stents, and tissue repair inserts (*Mousa, 2017a*).

### Nanomolecule type insights

Several forms of nanomolecules can be used in the healthcare industry. Quantum dots, nanoparticles, nanoshells, nanotubes, and nanodevices are examples of these compounds. Metal and metal oxide nanoparticles, liposomes, dendrimers, and conjugated polymers are examples of nanoparticles. The use of nanoparticles in various fields of healthcare is projected to accelerate development in the coming years (*SBWire, 2018*). The ability of these molecules to bind to chemical moieties in the form of scaffolds allows them to conduct a variety of functions. One of the most significant methods of nanotechnology is the use of nanoparticles for active targeting by attaching them to ligands of cell receptors. Because of their extensive use in diagnostics, metal and metal oxide nanoparticles are expected to be in high demand in this sector. It is expected that research to improve theranostics would affect industrial development (*Bawa et al., 2008*).

### Regional insights

North America led the nanomedicine industry with a sales share of more than 42 percent (Fig. 2) due to the presence of exponentially growing partnerships between established businesses in the field and nanomedicine startup organizations. Furthermore, government support, along with expanded R&D spending, accounts for the area's largest share of industrial space (*Soares et al., 2018*). The details of nanoparticles approved by FDA (Table 3) with their use in treatment of different disease. A variety of nanoformulations with pharmaceutical application are commercially available (Table 4).

## PUBLIC ACCEPTANCE OF NANOMEDICINE: A PERSONAL PERSPECTIVE

A lack of knowledge of a subject, results in narrow interpretations, including uninformed biases and comparisons. Prior biotechnologies, in the area of nanotechnology, have damaged the general understanding of nanotechnology solely by affiliation. Although societal bias is sluggish to change, it is possible that medical applications of nanotechnology would spark renewed interest and confidence in the sector by allowing people to live longer lives (*Grand Market Research, 2017*).

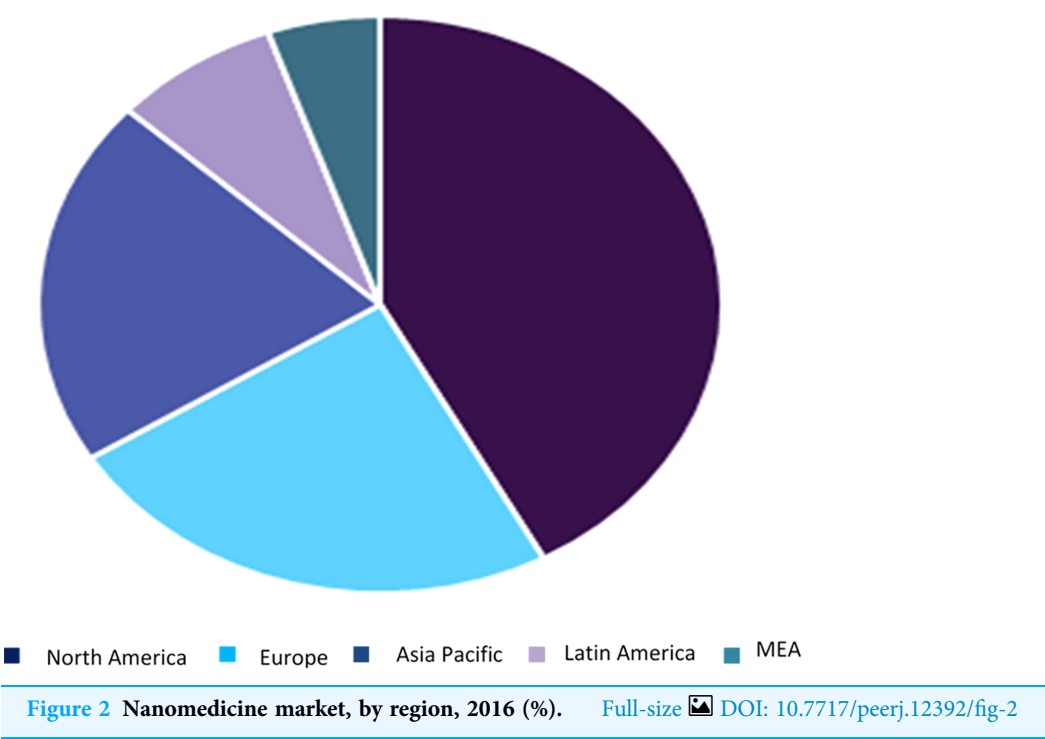

**Figure 2  Nanomedicine market, by region, 2016 (%).** 

## CHALLENGES AND OPPORTUNITIES IN FUTURE NANOMEDICINE: AN INDUSTRY PERSPECTIVE

If somehow the scope of nanotechnology technologies in medicine expands, it is critical to consider and advance contributions relevant to public health at the same time. A vast variety of nanomedicine advances have the potential to affect almost any medical specialty and reveal new approaches to increase the quality and length of life–these gains can be evaluated at both the human and community levels (*Berube, 2009*).

The unusual characteristics and phenomena that manifest due to their limited scale make nanotechnology applications appealing. The most commonly used scale for nanotechnology is 1–100 nm (*Pautler & Brenner, 2010*).

Engineering materials of this scale enables new medicinal treatments such as the development of nanoparticle-based medications with increased sensitivity, resulting in fewer adverse effects for patients (http://www.nano.gov).

Global research is currently ongoing to establish nanotechnology applications in cardiology, neurology, and a variety of other medical specialties (*Bawarski et al., 2008*).

Nanomedicines have been studied for their potential use in the targeted delivery of drugs to combat a wide range of diseases. This market viewpoint focuses solely on oncology-based nanomedicine therapeutics, which attract roughly two-thirds of study attention (*Kostarelos, Bianco & Prato, 2009*).

*Etheridge et al. (2013)* focus on public health interests which is critical to effectively advancing the discipline and evolving medical technology for clinical use. Simultaneously, information deficits about the fate and transport of engineered nanoparticles in biological systems must be resolved continuously (*Etheridge et al., 2013*).

**Table 3 Details of nanoparticles approved by FDA with their use in treatment of different disease.**

| S. No. | Nanoparticles type/drug | Application |
|---|---|---|
| 1. | Iron dextran colloid | Iron deficient anemia (*Anselmo & Mitragotri, 2019*) |
| 2. | Liposomal verteporfin | Mascular degeneration (*Anselmo & Mitragotri, 2019*) |
| 3. | Liposomal amphotericin B | Treatment and prevention of invasive fungal infection (*Anselmo & Mitragotri, 2019*) |
| 4. | Liposomal propofol | General anesthesia in open heart surgery or spinal surgery, morbidly obese patient (*Anselmo & Mitragotri, 2019*) |
| 5. | Perflutren lipid microspheres | Ultrasound contrast agent (*Anselmo & Mitragotri, 2019*) |
| 6. | Iron carboxymaltose colloid | Iron deficient anemia (*Anselmo & Mitragotri, 2019*) |
| 7. | Iron gluconate colloid | Iron deficient anemia (*Anselmo & Mitragotri, 2019*) |
| 8. | Iron sucrose colloid | Iron deficient anemia following autologous stem cell transplantation (*Anselmo & Mitragotri, 2019*) |
| 9. | Iron polyglucose carboxymethylether colloid | Iron deficient anemia following chronic kidney disease (*Anselmo & Mitragotri, 2019*) |
| 10. | Liposomal doxorubicin | Solid malignancies, pancreatic, breast, ovarian, leukemia, prostate, liver and metastatic cancer, Kaposi's sarcoma myeloma (*Anselmo & Mitragotri, 2019*) |
| 11. | Liposomal daunorubicin | Various leukemias, Kaposi's sarcoma (*Anselmo & Mitragotri, 2019*; *Obeid et al., 2018*) |
| 12. | Albumin-particle bound paclitaxel | Solid malignancies, lymphomas, breast, bladder, lung, neck and head, pancreatic, ovarian, prostate, liver and melanoma (*Anselmo & Mitragotri, 2019*) |
| 13. | Liposomal vincristin | Brain, lymphomas, melanoma and leukemia (*Anselmo & Mitragotri, 2019*), (*Obeid et al., 2018*) |
| 14. | Liposomal irinotecan | Solid malignancies, pancreatic, breast, brain or sarcomas cancer (*Anselmo & Mitragotri, 2019*) |
| 11. | Cytarabine: daunorubicin liposomal formulation | Different leukemias (*Anselmo & Mitragotri, 2019*) |
| 12. | RNAi lipid protein | Transthyretin mediated amyloidosis (*Anselmo & Mitragotri, 2019*) |
| 13. | Albumin bound Paclitaxel | Metastatic breast cancer, lung cancer (*Obeid et al., 2018*; *Desai et al., 2006*) |
| 14. | Liposomal cytarabine | Malignant lymphomatous meningitis (*Obeid et al., 2018*; *Desai et al., 2006*) |
| 15. | Paclitaxel polymeric micelle | Cell lung and breast cancer (*Obeid et al., 2018*; *Desai et al., 2006*) |

In view of the evidence mentioned above, nanotechnology has significant applications in healthcare management. Several nano-therapeutics have been licensed by the FDA for the treatment of hepatitis, cancer, cardiovascular disease, neurological disease, autoimmune disease, diabetes, elevated cholesterol, Parkinson's disease, and some infectious diseases during the last two decades (*Oberdorster et al., 2005*).

Hundreds of nano carrier-based drugs are still currently in different stages of preclinical and clinical production (http://nano.cancer.gov).

The use of nanotechnology in the area of medicine has already resulted in several breakthroughs and is rapidly progressing toward being a critical component of the healthcare system. This is shown by the growing number of scientific publications and patents awarded in the area of nanocarrier systems for drug delivery (*Etheridge et al., 2013*).

The transition of scientific discovery into marketable goods remains a problem, and the future of developing nanotherapeutics is dependent on turning promising research findings into profitable commercial technologies for stakeholders (*Wang et al., 2017b*). Nanotechnology has the potential to significantly improve precision medicine soon

**Table 4 Nanoformulations marketed product with their pharmaceutical application.**

| S. No. | Nanoformulations marketed products | Pharmaceutical application |
|---|---|---|
| 1. | Rapamune® | Immunosuppressant (*Kalepu & Nekkanti, 2015*) |
| 2. | Tricor® | Hypercholesterolemia (*Kalepu & Nekkanti, 2015*) |
| 3. | Emend® | Anti-emetic (*Kalepu & Nekkanti, 2015*; *Bharali et al., 2011*) |
| 4. | Megace ES® | Anti-anorexic (*Kalepu & Nekkanti, 2015*) |
| 5. | Avinza® | Phychostimulant drug (*Kalepu & Nekkanti, 2015*) |
| 6. | Triglide® | Hypercholesterolemia (*Kalepu & Nekkanti, 2015*) |
| 7. | Focalin® | Attention deficit hyperactivity disorder (*Kalepu & Nekkanti, 2015*) |
| 8. | Ritalin® | CNS stimulant (*Kalepu & Nekkanti, 2015*) |
| 9. | Daunoxome® | Anti-cancer (*Lohcharoenkal et al., 2014*; *Bharali et al., 2011*) |
| 10. | Myocet® | Anti-cancer |
| 11. | Doxil® | Anti-cancer, Antiretroviral (*Lohcharoenkal et al., 2014*; *Bharali et al., 2011*) |
| 12. | Caelyx® | Anti-cancer (*Lohcharoenkal et al., 2014*) |
| 13. | Genexol-PM® | Anti-cancer (*Lohcharoenkal et al., 2014*) |
| 14. | Transdrug® | Anti-cancer (*Lohcharoenkal et al., 2014*) |
| 15. | Abraxane® | Anti-cancer (*Lohcharoenkal et al., 2014*; *Bharali et al., 2011*) |
| 16. | Oncaspar® | Anti-cancer (*Lohcharoenkal et al., 2014*; *Bharali et al., 2011*) |
| 17. | Combidex® | Anti-cancer (*Bharali et al., 2011*) |
| 18. | Endorem® | MRI contrast agents (*Maximilien et al., 2015*) |
| 19. | Feridex® | MRI contrast agents (*Maximilien et al., 2015*) |
| 20. | Ontak® | Non-Hodgkin lymphoma (*Maximilien et al., 2015*) |
| 21. | Feraheme® | Anti-anaemic (*Maximilien et al., 2015*) |
| 22. | Marqibo® | Anti-cancer (*Maximilien et al., 2015*) |
| 23. | Docetaxel PNP® | Anti-tumor (*Maximilien et al., 2015*) |

(*National Nanotechnology Initiative, 2021*). Nano-diagnostics should be used more extensively to increase the sensitivity and reliability of diagnostic tests. Newer and more accurate nanoparticles may be detected, allowing us to make more precise disease diagnoses and management decisions (http://nano.cancer.gov).

## CONCLUSIONS AND FUTURE PERSPECTIVES

Novel nanotechnology-enabled systems show a promising approach to improving clinical possibilities and providing many benefits that are absent in existing therapeutic methods. In this review, we discussed an effort to highlight previous studies, patents, clinical investigations in the last 10 years, industrial utilities, and to attract medical attention to the potential prospects of nanomedicines for their promising therapeutic efficacies. The presence of patented nanomedicine would drive the Asia Pacific demand for nanomedicine (*Ragelle et al., 2017*). Future nanomedicine's challenges and opportunities: A detailed business perspective has been explored. The use of nanomedicines as a medication therapy in the treatment of global diseases such as cancer, malaria, HIV, and tuberculosis has been highlighted. Further studies in this new field can be guided by multidisciplinary techniques along with industry-academia collaboration. Until now,

translating experimental progress into marketable nano-products has been difficult, and the success of new nanomedicines is dependent on turning promising research results into profitable industrial technologies for stakeholders. Nanotechnology-based nano-systems can contribute to significant advances in personalized medicine in the immediate future. Nano-diagnostics should be used more extensively to increase the sensitivity and reliability of diagnostic tests. Newer and more accurate nanomedicines may be detected, leading to more precise cancer detection and treatment. Finally, the advancements and unusual characteristics outlined herein can pave the way for modern, highly efficient, and essential nanosystems.

## ACKNOWLEDGEMENTS

All the authors of this manuscript are thankful to their respective Departments/ Universities for successful completion of this study.

## ABBREVIATIONS

| | |
|---|---|
| TGA | Therapeutic Goods Administration |
| API | Active Pharmaceutical Ingredient |
| NCEs | New Chemical Entities |
| QbD | Quality by Design |
| PAT | Process Analytical Technology |
| ICH | International Conference on Harmonization |
| USFDA | United State Food & Drug Administration |
| EMA | European Medicines Agency |
| MAA | Marketing Authorization Application |
| FDA | Food & Drug Administration |
| PDMA | Pharmaceutical and Medical Device Agency |
| MHLW | Ministry of Health, Labor and Welfare |
| QALEYs | Quality Adjusted Life Expectancy Years |
| HTA | Health Technology Assessment |
| NBCDs | Non-Biological Complex Drugs |
| ITF | Innovation Task Force |
| TI Pharma | Top Institute Pharma |
| AML | Acute Myeloid Leukaemia |
| NDA | New Drug Application |
| BLA | Biologics Licence Application |
| RBCs | Red Blood Cells |
| $C_{max}$ | Concentration Maximum |
| R&D | Research & Development |
| ROS | Reactive Oxygen Species |
| AZTTP | 3 Azido-2,3-dideoxythymidine-5-triphosphate |
| BBB | Blood Brain Barrier |

### Funding

The authors received no funding for this work.

### Competing Interests

The authors declare that they have no competing interests.

### Author Contributions

- Rishabha Malviya conceived and designed the experiments, performed the experiments, analyzed the data, prepared figures and/or tables, authored or reviewed drafts of the paper, and approved the final draft.
- Shivkanya Fuloria conceived and designed the experiments, performed the experiments, analyzed the data, prepared figures and/or tables, authored or reviewed drafts of the paper, and approved the final draft.
- Swati Verma conceived and designed the experiments, performed the experiments, analyzed the data, prepared figures and/or tables, authored or reviewed drafts of the paper, and approved the final draft.
- Vetriselvan Subramaniyan conceived and designed the experiments, performed the experiments, analyzed the data, prepared figures and/or tables, authored or reviewed drafts of the paper, and approved the final draft.
- Kathiresan V. Sathasivam conceived and designed the experiments, performed the experiments, analyzed the data, authored or reviewed drafts of the paper, and approved the final draft.
- Vinoth Kumarasamy conceived and designed the experiments, performed the experiments, analyzed the data, authored or reviewed drafts of the paper, and approved the final draft.
- Darnal Hari Kumar conceived and designed the experiments, performed the experiments, analyzed the data, authored or reviewed drafts of the paper, and approved the final draft.
- Shalini Vellasamy conceived and designed the experiments, performed the experiments, analyzed the data, authored or reviewed drafts of the paper, and approved the final draft.
- Dhanalekshmi Unnikrishnan Meenakshi conceived and designed the experiments, performed the experiments, analyzed the data, authored or reviewed drafts of the paper, and approved the final draft.
- Shikha Yadav conceived and designed the experiments, performed the experiments, analyzed the data, prepared figures and/or tables, authored or reviewed drafts of the paper, and approved the final draft.
- Akanksha Sharma conceived and designed the experiments, performed the experiments, analyzed the data, prepared figures and/or tables, authored or reviewed drafts of the paper, and approved the final draft.

- Neeraj Kumar Fuloria conceived and designed the experiments, performed the experiments, analyzed the data, prepared figures and/or tables, authored or reviewed drafts of the paper, and approved the final draft.

## Data Availability

This is a review article, so there is no raw data.

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
