# Peer review of "Commercial utilities and future perspective of nanomedicines"

_PeerJ, doi:10.7717/peerj.12392_

## Round 0.1 · original submission · Major Revisions

Please address the concerns of both reviewers, paying close attention to the comments and suggestions of reviewer #1

·

Basic reporting

no comment.

Experimental design

In the present study, the authors tried to collect material on the importance of the field of nanomedicine. Although the topic is very broad, in general, materials have been collected in some formats such as the application of nanomedicine, economic aspects, as well as challenges and future perspectives.

Validity of the findings

This study has many problems in terms of writing. My advice is that if it is to be published, it must first be greatly modified in writing. The next thing that worries me is the generality of the content. Although the topic has been selected as a whole, looking at each section, we see that no specific information is provided to the reader. For example, the part that talks about Taiwan's strategy is very unnecessary and vague. On the other hand, the Standards and Guidelines section does not contain detailed information. Content generally does not add anything new to the reader. As another example, in the Clinical Trials section, it is not stated how long these phases last for drugs derived from nanoparticle technology.
Another concern that cannot be ignored is the inconsistency of references with the content. For example, there is no connection between references “Blanco E, Hen H, Ferrari M. 2018” and “Brigger I, Dubernet C, and Couvreur P. 2002” with the referenced material.
This manuscript also requires tables that contain useful and up-to-date information such as nanoparticle classification (based on application or world usage or …), number of FDA-approved nano-drugs, and diseases that are currently treatable with nanoparticles.
Here are some grammatical errors.
Line 37, “was” must change with “were”.
Line 43, “precise” must change with “precision”.
Line 50, “offers” should be replaced with “offer”.
Line 51, “in case” must change with “in the case”.
Line 97, “has” must change with “have”.
Lines 108-112, the sentence (The progress discussed for…) is very long and confusing. Please amend it.
Line 116, “that” should be deleted.
Line 151, “or identifying them from”.
Line 199, “to increased” should be replaced with “to increase”.
Line 151, “steps”.
Line 392, “results”.
Line 539, “phenomena”

Additional comments

Providing more detailed data, eliminating some unnecessary parts such as Taiwan strategy, providing a rich table of contents, and synchronizing references with the content provided are important operations that should be considered.

Reviewer 2 ·

Basic reporting

This is a review on nanomedicine commercialization, but it seemed unfocused and the presented information appeared arbitrary. Few commercialization details are provided. The first part focused on regulatory frameworks for nanomedicine, but didn't include much concrete information. The next part discussed basics of nanomedicine, but again without depth and seemingly arbitrary manner. The selection of table 1 which focuses on Taiwanese nanomedicine and table 2 which lists seemingly random patent literature also makes this work unfit for publication.

Experimental design

NA

Validity of the findings

NA

Additional comments

NA

---

## Round 0.2 · Minor Revisions

Please address a remaining concern of the reviewer and provide a list of abbreviations used in the manuscript.

·

Basic reporting

Thank you to the authors for providing all corrections.

Experimental design

No comment

Validity of the findings

As far as my knowledge, the manuscript was well written and had no grammar or spelling errors. However, if the Editor finds this necessary, it would be very great if the authors provide a list of abbreviations used in the body of the texts.

---

## Round 0.3 · accepted · Accept

Thank you for addressing the remaining concerns and for the amending your manuscript accordingly. The revised version of this manuscript is acceptable now.